# The Use of Camera Traps and Auxiliary Satellite Telemetry to Estimate Jaguar Population Density in Northwestern Costa Rica

**DOI:** 10.3390/ani12192544

**Published:** 2022-09-23

**Authors:** Víctor H. Montalvo, Carolina Sáenz-Bolaños, Juan C. Cruz-Díaz, Eduardo Carrillo, Todd K. Fuller

**Affiliations:** 1Instituto Internacional en Conservación y Manejo de Vida Silvestre, Universidad Nacional, Heredia 1350-3000, Costa Rica; 2Department of Environmental Conservation, University of Massachusetts, Amherst, MA 01003, USA; 3Namá Conservation, Heredia 40101, Costa Rica; 4Amazon Conservation Team, Arlington, VA 22203, USA

**Keywords:** density, dry forest, *Panthera onca*, spatial capture-recapture, sex ratio

## Abstract

**Simple Summary:**

We used data from a grid of camera traps, combined with satellite telemetry data from one female jaguar, to estimate jaguar population density in northwestern Costa Rica. Our estimate of 2.6 females and 5.0 males per 100 km^2^ was influenced by camera placement and sex of the jaguar, and indicated the importance of telemetry data to improve field design and parameter estimation. We recommend population assessments of at least 3 consecutive months, expanding the survey area to one several times the size of an individual’s home range, and including sex and camera placement considerations to reduce bias in jaguar density estimates.

**Abstract:**

Regular evaluation of jaguar (*Panthera onca*) population status is an important part of conservation decision-making. Currently, camera trapping has become the standard method used to estimate jaguar abundance and demographic parameters, though evidence has shown the potential for sex ratio biases and density overestimates. In this study, we used camera trap data combined with satellite telemetry data from one female jaguar to estimate jaguar population density in the dry forest of Santa Rosa National Park in the Guanacaste Conservation Area of northwestern Costa Rica. We analyzed camera trap data collected from June 2016 to June 2017 using spatial capture-recapture methods to estimate jaguar density. In total, 19 individual jaguars were detected (11 males; 8 females) with a resulting estimated population density of 2.6 females (95% [CI] 1.7–4.0) and 5.0 male (95% [CI] 3.4–7.4) per 100 km^2^. Based on telemetry and camera trap data, camera placement might bias individual detections by sex and thus overall density estimates. We recommend population assessments be made at several consecutive 3-month intervals, that larger areas be covered so as not to restrict surveys to one or two individual home ranges, as in our case, and to carry out long-term camera monitoring programs instead of short-term studies to better understand the local population, using auxiliary telemetry data to adjust field designs and density estimations to improve support for jaguar conservation strategies.

## 1. Introduction

Most carnivores are elusive and solitary species [1], thus monitoring such difficult-to-detect species is a challenge to answering ecological questions. The estimation of population parameters of endangered species is crucial to understand their ecology and distribution [2,3], thus appropriate conservation strategies required accurate and trustworthy information [4,5]. Several non-invasive methods, such as DNA analysis of scats or hair, camera trapping, and acoustic assessments, allow “capturing” individuals with minimal or no handling stress [6,7,8], in contrast to other techniques that involve physical capturing; e.g., telemetry and other animal tagging [9,10]. Jaguars (*Panthera onca*) are the largest felid in the Neotropics [11] and a near-threatened species [12] roughly inhabiting 50% of their original historic range distribution [13]. Though jaguars play a key role in the ecosystem dynamics by balancing ecosystem services and ecological processes [1,14], their local populations are threatened and vulnerable [15]. Therefore, regular evaluations of local jaguar population status are an important part of conservation decision-making.

Due to the elusiveness and rarity of jaguars, field studies of their ecology and behavior are difficult [9,16]. Often, camera traps are recommended to study elusive mammals like tigers (*Panthera tigris*) and jaguars [3,6,17,18]. Currently camera traps have become a standard method commonly used to elucidate jaguar abundance and demographic parameters [3,6,8] using their distinctive and unique rosette patterns [6,7] with capture-recapture methods [8,19,20]. Although simultaneous comparison and adjustments of jaguar population estimates with satellite telemetry are limited [21,22], evidence has shown sex ratio biases and density overestimates derived from camera trap data [23]. Furthermore, scale bias due to the use of camera traps in small areas (<100 km^2^ [2]) hinders accurate density estimation. Previous capture-recapture (CR) jaguar density estimates indicated the overestimation of jaguar density by 70% when contrasting simultaneous satellite-telemetry tracking and camera trapping [21]; other studies showed a few discrepancies [22]. With novel approaches such spatial capture–recapture (SCR [8]), spatially explicit information allows us to estimate the centers of activity, thus making assumptions about home range shapes [24]. Testing and adjusting repeatedly camera trap methods at long-term research sites would allow for better monitoring of individuals and systems over time [25]. Thus, incorporating satellite telemetry parameters instead of movement estimates to derive camera detection probabilities can allow more robust inferences about space to provide accurate density estimates [26].

Here, we describe a jaguar population in the Santa Rosa Sector of Guanacaste National Park in the dry forest of northwestern Costa Rica using camera traps and spatial capture-recapture methods (SCR; [27]), along with satellite telemetry data from one female jaguar. We examined the relationship of trail and off-trail camera placement on population density estimates integrating movement parameters, as well as how the sex-bias incurred by camera placement might affect detection rates of individuals and thus estimates of population structure. We compare camera trap estimates of density with those derived from satellite telemetry data and make methodological recommendations to improve future jaguar population estimates.

## 2. Materials and Methods

### 2.1. Study Area

This study was conducted in the Santa Rosa sector of the Guanacaste Conservation Area located in northwest Costa Rica [10°53’01″ N 85°46’30″ W; 28]. Santa Rosa encompasses 387 km^2^ and is dominated by some of the last remaining tropical dry forests in Central America [28,29]. Average annual rainfall of 1600 mm is highly seasonal (monthly averages from 0 mm to 1040 mm). The wet season (months with ≥40 mm of rain) is May to November, and the dry season (with almost no rain and temperatures over 37 °C) is December to April. Due to the rarity of dry forest ecosystems, a large-scale restoration effort was initiated in the 1980s involving protected area status, the recovery of abandoned pastures by active fire suppression [30], protection from many human activities, and the recovering of large vertebrate populations. In Santa Rosa there are two important sea turtle nesting beaches: Playa Nancite (length = 1.05 km) where massive numbers of turtles come ashore during the wet season [31]; Playa Naranjo (length = 5.64 km) where turtle nesting occurs year-round but increases during the wet months [32]. As an important prey item, turtle nesting peaks influence the movement and behavior of jaguars in the area [33].

### 2.2. Data Collection

From 15 June 2016 to 13 June 2017, we conducted a constant camera trap effort (trap nights) in Santa Rosa. Fifty-eight automatic trail cameras (Bushnell^®^, Trophy Cam models 119,436, 119,446, 119,456) were deployed in 29 hexagons in a grid array of 3 km^2^ each (Figure 1A,B). Half of the cameras (one camera per site) were at a trail location that jaguars were likely to use, and the other half at an off-trail location within 200 m of each hexagon centroid; on average each off-trail location was 0.59 km ± 0.25 SD away from available trails (Figure 1A,B), and the total camera array covered an area of 87 km^2^.

Each camera was affixed to a tree at a height of ~40 cm and set to be active for 24 h/day in photo mode with a minimum delay of 1 sec between consecutive triggers. Once deployed, cameras were checked on average every month to replace batteries and change SD memory cards, if necessary. For each camera deployment, we recorded the location and camera operation dates.

We identified jaguars based on individual pelage (rosette and spot) patterns [6]. Since we only used one camera per station to identify jaguar individuals, and to avoid identification problems from only getting photos from one side/flank, a previous long-term jaguar identification catalog for the area with different photos of body sides and angles was used to help maximize identifications. Photos were independently identified by 3 investigators and results were compared to minimize errors. We also classified jaguar sex (male, female, unknown), age (cub, young, adult), and whether individuals were collared or not collared. Adults were sexed by presence/absence of testicles and nipples [20] and aged by their size and physical appearance to categories of cubs (<12 m), young (12–24 m), and adults (>24 months; [20]).

For satellite telemetry data collection, we immobilized an adult female jaguar, caught with a foot-snare trap with a combination of 5 mg/kg of ketamine (10% ketamine, Bremer Pharma GmbH, Warburg, Germany) mixed with 2 mg/kg xylazine (Procin Equus 10%, Pisa Agropecuaria). The 42-kg adult female was medically evaluated by a wildlife veterinarian and fitted with a GPS collar (Lotek Engineering, Newmarket, ON, Canada) programmed to record the jaguar’s position every 2 h. Handling and capture protocols followed guidelines of the American Society of Mammologists [34] and were approved by the Environmental Minister of Costa Rica (ACG-PI-034–2014), following the ethics and research procedure guidelines of the National University. We used the location data to estimate the size of the area traversed by the jaguar (Table 1; Jaguar01) with a kernel density estimate [35,36].

### 2.3. Density Estimation

For adult jaguar density estimates we used the package oSCR version 0.42 (Sutherland et al., 2019) in R version 3.3.2 [37]. The oSCR package is based on spatial capture models of *N* individuals associated with specific location patterns that represent the center of activity, as well as the specific probability of observing one individual relating to the distance from other individuals center of activity [24]. It also allows the building of models with class sex population information [38] and multiple seasons in the model’s structure [24]. In this study we used season, sex structure, and camera placement (trail/off-trail) to investigate their effects on population density (*D*), the baseline encounter rate (*p*), and space use (*sigma*) (Table 2). The area within the distribution of individual activity centers assumed to be randomly distributed is known as state space (*S*) and was created using a buffer area three times *sigma* (6000 m; based on telemetry movement data) around the camera array, with 0.5 × 0.5 km resolution [24]. Candidate models that represented hypotheses analyzed were evaluated using the Akaike Information Criterion corrected for small sample size (AICc) [39], and throughout model comparison we determined the most plausible models from AICc differences (ΔAICc) and weights (*W*). If a model included a single effect that did not reduce the AICc value compared with a null model (model response ~ 1), it was not considered as a supportive effect.

### 2.4. Use of Satellite Telemetry to Improve Density Estimation

To identify potential sources of bias between satellite telemetry and camera trap data that potentially affected population estimates, we used the dataset (*n* = 5924 locations) of the collared jaguar female. To improve the density estimation, we used space use parameters from satellite telemetry, to adjust “*S”* and “*sigma*” parameters. Additionally, descriptive statistics from camera trap and telemetry data within this array were used to depict the population structure; monthly number of different jaguars detected, comparison of satellite telemetry locations within multiple nested buffer distancing ratios (50, 100, 150, 200 and 250 m) around cameras deployed at trail and off trail locations, and spatial locations of the collared female detected within the density grid array.

## 3. Results

### 3.1. Camera Trapping and Individual Detection

A total effort of 18,170 continuous trap-nights yielded 948 identifiable jaguar photos, resulting in 188 independent identifiable jaguar photo captures, and 19 different jaguar individuals (females = 8, males = 11). Camera trap efforts were constant across sampling seasons (Table 1), recording average 1.2 independent jaguar photo captures/100 trap nights. The total number of jaguar captures registered was frequently high (91%) at trail locations (Table 1) compared to off trail locations (9%), detecting both jaguar females (Figure 1A) and males (Figure 1B) mostly near (<1 km) the coast line. The accumulated number of different jaguar individuals across sampling days reported more jaguar individuals at trail camera locations; however, the number of males was high (Figure 2) compared to females, and for such off-trail locations jaguar individuals were registered less often than at trail camera locations; nevertheless, female individuals there were recorded more frequently than males. Monthly records of jaguar individuals were relatively constant during the sampling effort (*n*= 6) except for June (Figure 3).

### 3.2. Density Estimation

Model selection based on AIC (Table 3) showed as the top model the one assuming constant density (*D*), encounter rates (*p*) that varied by sex and camera location, as well as specific sex and session on space use (*Sigma*) (AIC:2526, *w*: 0.98). Maximum likelihood parameter estimates of the top model showed a density of 7.6 (95% [CI] 5.1–11.5) jaguars per 100 km^2^ (applied to the buffered area of 160 km^2^), segregated into 2.6 (95% [CI] 1.7–4.0) female jaguars per 100 km^2^ and 5.0 male jaguars (95% [CI] 3.4–7.4) per 100 km^2^ (Figure 4 and Figure 5), and additional probability of being a male (Ψ Prob) of 0.656 (Table 4).

Variation in baseline detection rates showed male jaguars at off-trail locations (*p* = 0.0003; 95%CI = 0.0001–0.001) were significantly lower than females (*p* = 0.002; 95%CI = 0.002–0.005). Overall, jaguar baseline detection rates were significantly higher at trail locations than off-trail locations (Figure 5); nonetheless, female jaguars (0.0247; 95%CI = 0.009–0.0681) and male jaguars (0.004; 95%CI = 0.0007–0.0018) detection rates were not statistically different at either location type. Estimated average spatial scale parameter (*sigma*) was 2102 m (95%CI = 1691.2–2617.6) and showed unequal space use; male jaguars use was greater than that of female jaguars, with some variation across sessions (Figure 6).

### 3.3. Data and Results Comparisons

Our camera trap array embraced almost 95% of the home range of the collared female jaguar, but her image was recorded at only 13 camera locations (Figure 7), mostly near the coastline and almost none at cameras in the more northernly part of her range, especially where she clearly spent time (more often during the non-peak sea turtle nesting season [36]). Comparison of satellite telemetry locations within multiple nested buffer distancing ratios around cameras deployed at trail and off-trail locations showed a high number of accumulated locations at trail camera deployments (Figure 8), but she still was not photographed at those cameras in the north. We note that few other identified jaguars were photographed in that area, as well (Figure 1).

Furthermore, photos of at least 1 other female and 1 or more males were recorded within the collared female’s range in each month (Figure 3) and, in fact, an average of 6 different individuals per month were identified in the study area, except in June. Given that camera arrays such as ours likely do not record all individuals in an area over a period of time [40], this number of individuals in the total camera array of 87 km^2^ is similar to our estimated density (7.6/100 km^2^).

## 4. Discussion

This study provides a fine-scale, robust jaguar population density estimate, considering the methodological constraints of site placement and sex bias, by incorporating camera trap results with data from one collared female jaguar in the tropical dry forest ecosystem.

Jaguar population estimates that address the effect of detectability and sample size are numerous [41]; however, few density studies delve further in bias linked to detection as result of individual sex or camera location. For example, Maffei et al. [40,42] found male jaguars are associated with wide trails as easily accessible travel routes, whereas female jaguars use both trails and dense forest areas the same, hypothesizing that dense forest provides alternative travel routes to avoid cub infanticide by dominant males. For tigers in India, a similar pattern was identified in density studies; depending on sex and age, photo rates decreased or increased, assuming old well-established tigers moved freely, and submissive individuals avoid encountering them [43]. Our findings indicated high numbers of male jaguars on trails, different than females who used off trail locations more often; this is the same pattern observed in Venezuela in a year-round jaguar density study where females with cubs avoided places highly frequented by unrelated individuals [20].

Jaguar density estimates did not fluctuate significantly across four seasons during the sample year; therefore, we report an average density estimated of 7.6 jaguars/100 km^2^. Previous jaguar estimates in Santa Rosa reported 2.23 jaguars/100 km^2^, using non-spatially explicit methods [44], whereas other studies did not register enough individual records to perform CR models (i.e., recorded only two juvenile males and two females; [45]). Compared with prior efforts, current jaguar population numbers at Santa Rosa showed a relative high density, presumably because of the recovery of prey populations, as well as the massive availability of sea turtles at most Santa Rosa beaches were “arribadas” occur [26,41], and sea turtles are a significant low-cost reward [33,46] that provide an important food source for this jaguar population.

With regard sex-specific jaguar density, we found differences for males (mean= 5.0 jaguars/100 km^2^) and females (mean = 2.6 jaguars/100 km^2^), a pattern previously reported in high density areas in South America [4,39]. The baseline encounter rates for jaguar males and females at trail and off-trail locations showed that, though female jaguars were less abundant, they are more likely to be photographed at both camera placements. Jędrzejewski et al. [20] found jaguar females without offspring are less shy and likely to visit the same places as males. Additionally, the findings obtained in this study are consistent with other taxa where camera location placement influenced photo rates results, as well as species detection [47,48,49], highlighting strong methodological constraints as a result of ignored behavior patterns. Jaguar males seem to walk longer distances than jaguar females based on camera trap data, similar to what Morato et al. [10] found for regional data movement analysis; jaguar males tend to use larger areas than females.

Telemetry home range data of a collared female identified intense space use that almost fit our camera array area. Despite this, the female used trail locations the most, and thus camera placement at trail locations could significantly increase the detection chances of this collared female. Though camera site placement at trail locations might shade patterns of distribution or intra-specific interactions, the use of camera placement at trail locations could improve detection of individuals as CR field arrays [6,17]. Within the home range of our collared female a consistent number of individuals (mean = 6) was detected by our camera array each month, suggesting different individuals occasionally overlap home ranges during the year. This potentially could affect the detection of individuals for population estimates because some individuals may temporally use or avoid specific areas as long as territorial individuals are present [21,22,26].

These findings suggest that camera location arrangement might influence results in highly seasonal ecosystems, especially for estimates that do not account for sex and camera placement as covariates, resulting in biased estimates. Though most camera trap studies ignore the effects of camera placement on estimates (abundance, population index and richness), animal distribution and movements follow non-random patterns; therefore, standardizing and classifying placement sites regardless of the ecosystem is important, and these finding can be extrapolated to other ecosystems using camera trapping in conservation studies.

## 5. Conclusions

Our results support the use of SCR as a robust method to estimate jaguar populations if the frequency of occurrence of jaguar individuals is high enough. For long-term study sites, we recommended gathering a spatial understanding of individual movements by incorporating satellite telemetry parameters to adjust model parameters, as well as considering camera arrangements that could provide more accurate abundance estimates. Our estimates from Santa Rosa suggest that the jaguar population might have increased in recent years, identifying it as an important jaguar conservation hotspot in Costa Rica. Based on our detection rates, further jaguar population estimates at Santa Rosa should occur in time periods >3 months, and camera coverage should be of larger areas that do not restrict the study to one or two individual home ranges.

## Figures and Tables

**Figure 1 animals-12-02544-f001:**
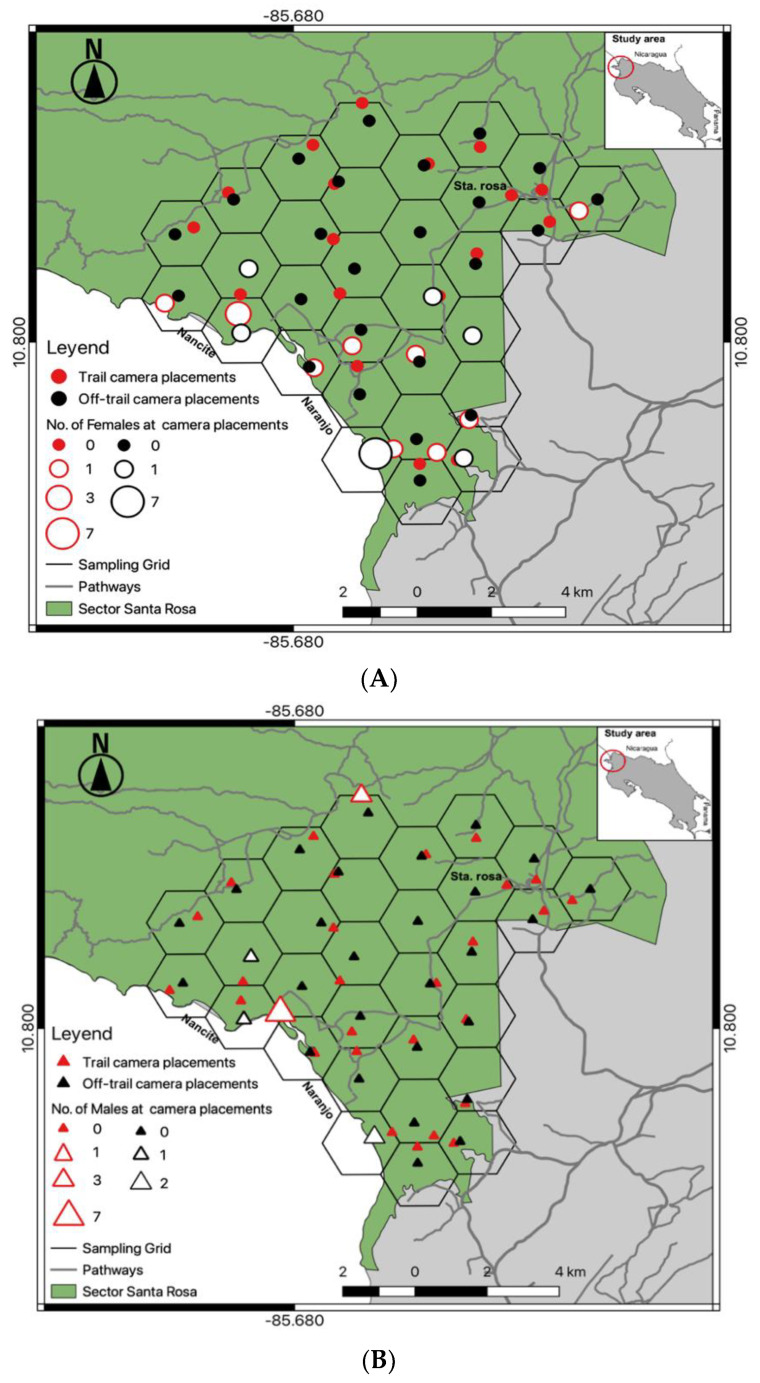
Camera trap deployment array at trail (*n* = 29) and off-trail (*n* = 29) locations in Sector Santa Rosa, Guanacaste Conservation Area, Northwestern Costa Rica; (**A**) Spatial detections of different female (♀) jaguars at camera locations; (**B**) Spatial detections of different male (♂) jaguars at camera locations.

**Figure 2 animals-12-02544-f002:**
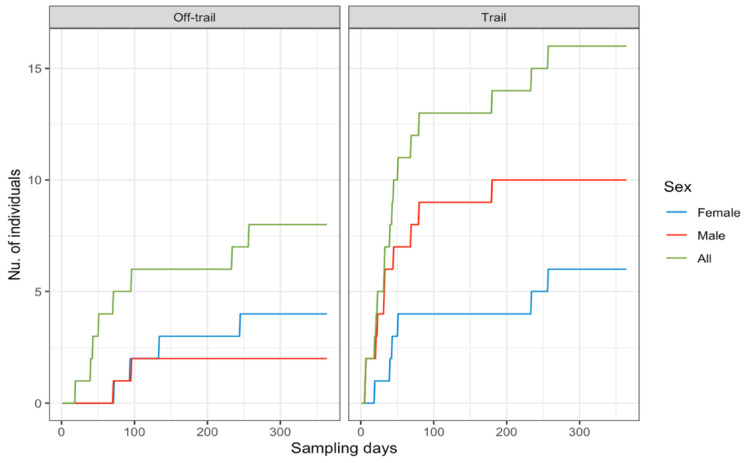
Accumulated number of jaguar individuals by sex at trail/off trail camera placement locations across sampling days for a jaguar camera trap density study in Sector Santa Rosa, Guanacaste Conservation Area, Northwestern Costa Rica.

**Figure 3 animals-12-02544-f003:**
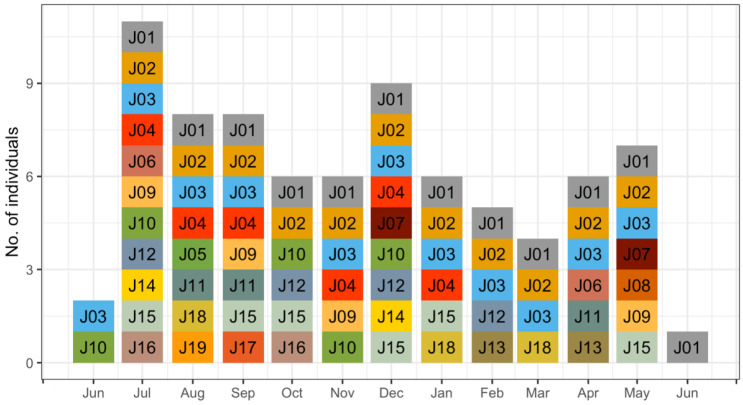
Monthly number of jaguar individuals (number- and color-coded) registered (June 2016–June 2017) in a camera trap density study in Sector Santa Rosa, Guanacaste Conservation Area, Northwestern Costa Rica.

**Figure 4 animals-12-02544-f004:**
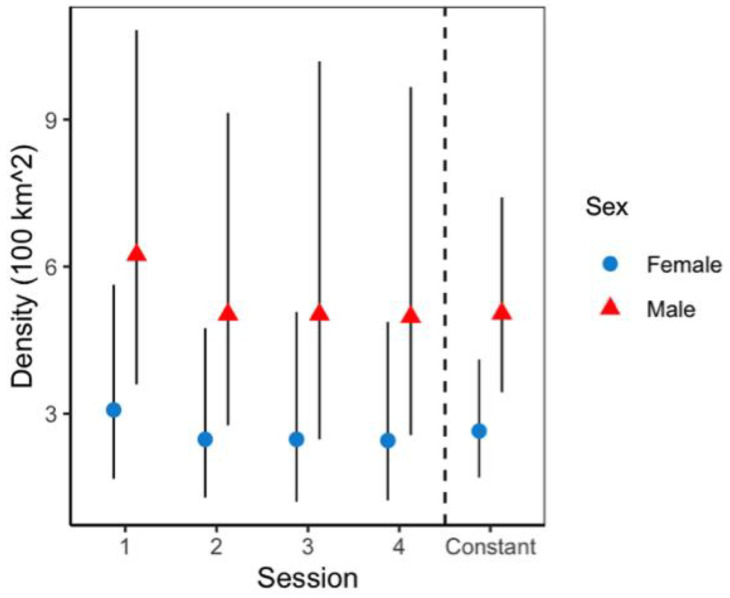
Sex/session specific jaguar density, from top model structure in Sector Santa Rosa, Guanacaste Conservation Area, Northwestern Costa Rica. The black line represents 95% confidence intervals.

**Figure 5 animals-12-02544-f005:**
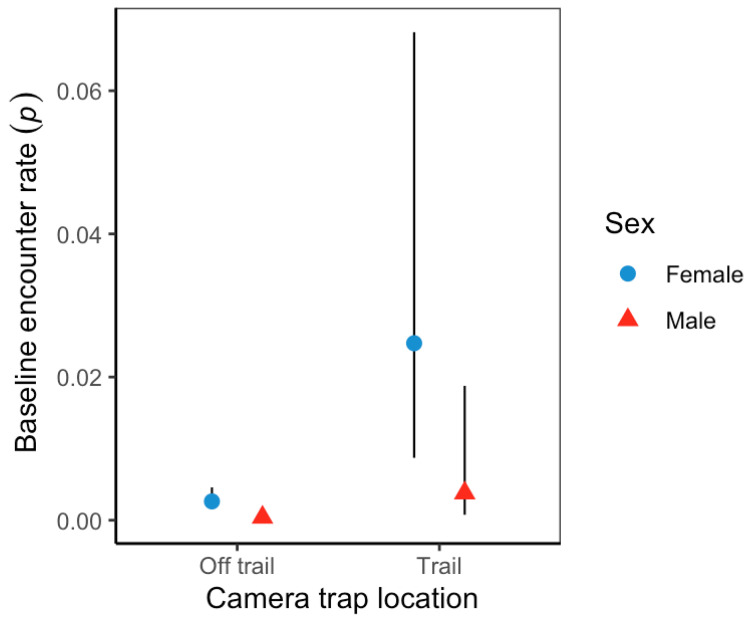
Sex-/camera trap location-specific effect on jaguar baseline encounter rates, from top model structure in Sector Santa Rosa, Guanacaste Conservation Area, Northwestern Costa Rica. The black line represents 95% confidence intervals.

**Figure 6 animals-12-02544-f006:**
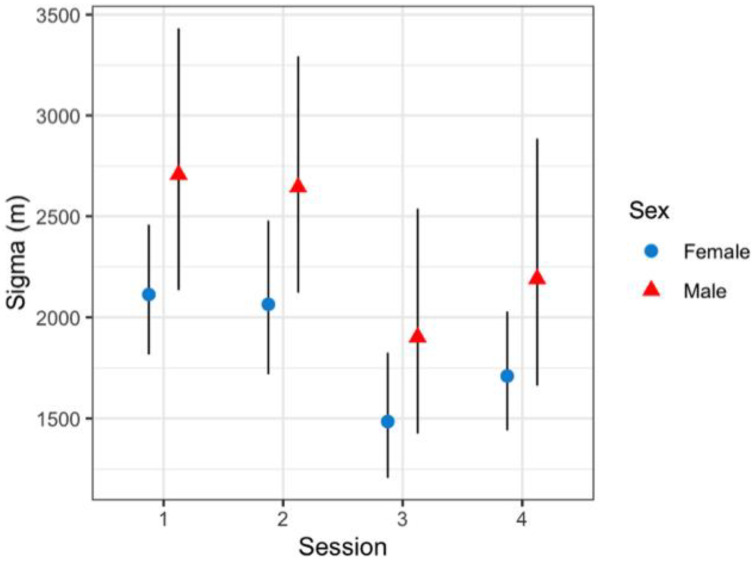
Sex-/season-specific effect on jaguar sigma (m) from top model structure in Sector Santa Rosa, Guanacaste Conservation Area, Northwestern Costa Rica. The black line represents 95% confidence intervals.

**Figure 7 animals-12-02544-f007:**
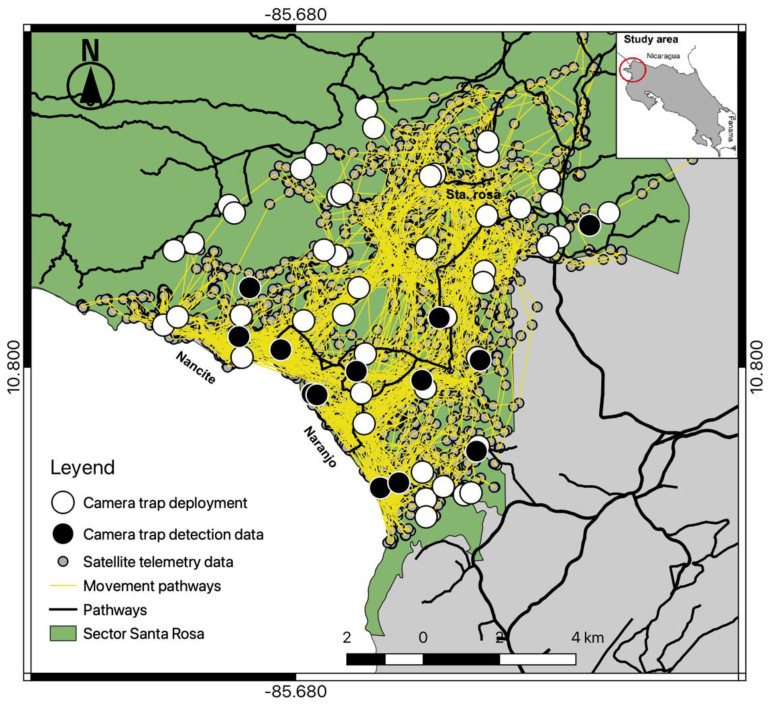
Spatial detections by camera trap of the radio-collared female jaguar within the study area, overlaid on the satellite telemetry data for the same individual in Sector Santa Rosa, Guanacaste Conservation Area, Northwestern Costa Rica.

**Figure 8 animals-12-02544-f008:**
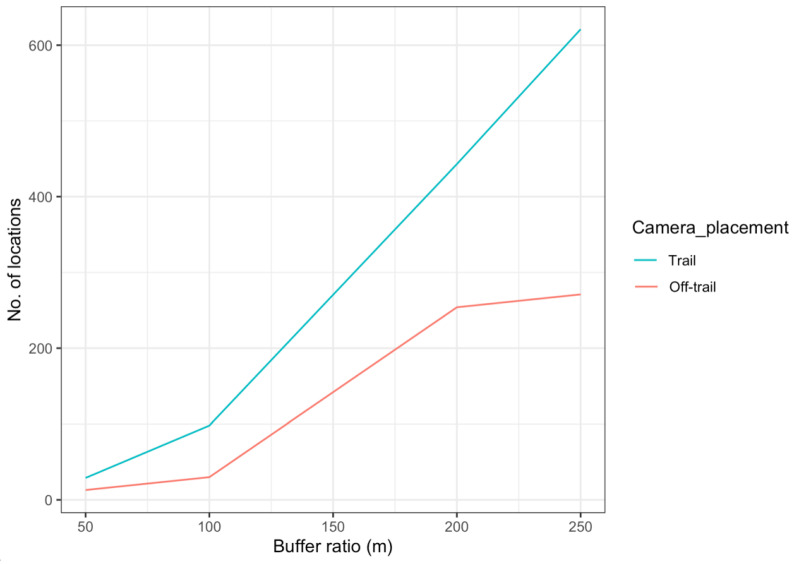
Number of satellite telemetry locations of a collared female jaguar, located within multiple buffer ratio distances around camera location placements (trail/off-trail) in Sector Santa Rosa, Guanacaste Conservation Area, Northwestern Costa Rica.

**Table 1 animals-12-02544-t001:** Individual jaguar captures registered at different sites (trail/off-trail) in Sector Santa Rosa, Guanacaste Conservation Area, Northwestern Costa Rica.

ID Individual	Sex	Camera Placement Location	
		Trail	Off-Trail
Jaguar01	F	59	11
Jaguar02	F	28	1
Jaguar04	F	8	---
Jaguar11	F	3	---
Jaguar13	F	2	---
Jaguar16	F	2	---
Jaguar08	F	1	---
Jaguar19	F	---	1
Jaguar03	M	29	---
Jaguar12	M	12	---
Jaguar10	M	7	---
Jaguar15	M	5	1
Jaguar09	M	4	---
Jaguar14	M	3	---
Jaguar18	M	---	3
Jaguar17	M	2	---
Jaguar05	M	2	---
Jaguar06	M	2	---
Jaguar07	M	2	---

**Table 2 animals-12-02544-t002:** Sampling effort for a jaguar camera trap density study in Sector Santa Rosa, Guanacaste Conservation Area, Costa Rica.

Session	Period	Trap Array (km^2^)	State Space (km^2^)	No. of Camera Stations			Trap Nights	No. of Occasions	No. of Indiv.	Average Cap.	Spatial Cap.
				Trail	Off-Trail	Total					
1	15 June–14 September 2016	87	160	29	29	58	4394	92	16	3.69	1.75
2	15 September–14 December 2016	87	160	29	27	56	4954	91	13	4	2.08
3	15 December–14 March 2017	87	160	28	27	55	4857	90	10	3.7	1.8
4	15 March–13 June 2017	87	160	28	27	55	3965	91	11	3.45	1.73

**Table 3 animals-12-02544-t003:** Model selection results for 11 candidate models analyzed including: session effects (session), male/female sex effect (sex), trail/off trail camera location (loc) and constant effect (~1), in Sector Santa Rosa, Guanacaste Conservation Area, Northwestern Costa Rica.

Density	Detection	Space Use	K	AIC	Delta AIC	Weight	Cum. Weight
D (~1)	p(~sex + loc)	sig(~session + sex)	10	2556	0	0.98	0.98
D (~session)	p(~sex + loc)	sig(~session)	12	2564	7.7	0.19	0.99
D (~1)	p(~sex + loc)	sig(~1)	6	2567	11.2	0.001	1
D (~session)	p(~loc)	sig(~sex)	9	2614	57.6	<0.001	1
D (~1)	p(~sex)	sig(~session)	8	2693	137.1	<0.001	1
D (~1)	p(~sex + session)	sig(~session)	11	2695	139.3	<0.001	1
D (~1)	p(~sex + session)	sig(~1)	8	2704	148.1	<0.001	1
D (~1)	p(~session)	sig(~session)	10	2739	183.1	<0.001	1
D (~1)	p(~1)	sig(~session)	7	2742	185.9	<0.001	1
D (~1)	p(~1)	sig(~1)	4	2747	190.6	<0.001	1
D (~1)	p(~session)	sig(~1)	7	2752	196.2	<0.001	1

**Table 4 animals-12-02544-t004:** Maximum likelihood parameter estimates from the top model of jaguar density, that included constant density D (~1), based line detection varied according to sex (sex) and trail/off trail camera location (loc), sex- and session-specific space use: sig (~session+ sex), and sex ratio Ψ, in Sector Santa Rosa, Guanacaste Conservation Area, Northwestern Costa Rica.

Parameter	Coefficient	SE
*p* (intercept: female, off trail)	−5.94	0.286
*p* (male)	−1.898	0.284
*p* (trail)	2.265	0.255
*sig* (intercept: female, session 1)	7.656	0.077
*sig* (session 2)	−0.024	0.099
*sig* (session 3)	−0.353	0.118
*sig* (session 4)	−0.212	0.103
*sig* (male)	0.248	0.125
Density	−2.565	0.15
Ψ Prob	0.646	0.297

## Data Availability

Not applicable.

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
