# Peer review of "The Use of Camera Traps and Auxiliary Satellite Telemetry to Estimate Jaguar Population Density in Northwestern Costa Rica"

_animals, 2022, doi:10.3390/ani12192544_

Round 1
Reviewer 1 Report
This paper presents camera traps and satellite telemetry to estimate the jaguar population structure in northwestern Costa Rica. The overall quality of the article is in line with the expectation. However, the authors may consider the following comments
1. Introduction
The reasons for the choice of camera traps and satellite telemetry should be stated. In addition, I suggest expressing the principal results of the investigation.
2. Materials and Methods
In section 2.2, the paper demonstrates how to collect camera trap data. However, the collection of Satellite telemetry data should also be proposed.
In describing the methods of investigation, you should direct readers to sufficient details so they can repeat the work. In the current version, I think it’s hard to repeat.
A combination of Satellite telemetry and camera trap seems a particular method; therefore, you should state how to combine them. Because they have different views on monitoring the ground, how to fuse data from different sources?
3. Results
I suggest that you should give an overall description of the experiments.
4. Discussion
I think the significance of the results is not discussed adequately. For example, you should discuss the relationship between the Camera trap and satellite telemetry data.
I suggest showing your results in contrast with previous works.
Author Response
Reviewer #1:
Introduction; The reasons for the choice of camera traps and satellite telemetry should be stated. In addition, I suggest expressing the principal results of the investigation.
- We add extra references and now emphasize why we used telemetry movement data in population estimates; Line 55-61.
- Materials and Methods; In section 2.2, the paper demonstrates how to collect camera trap data. However, the collection of Satellite telemetry data should also be proposed.
- We have added a description of satellite telemetry data collection; Line 120-130.
In describing the methods of investigation, you should direct readers to sufficient details so they can repeat the work. In the current version, I think it’s hard to repeat.
- We elaborated and detailed our methods throughout the section.
A combination of Satellite telemetry and camera trap seems a particular method; therefore, you should state how to combine them. Because they have different views on monitoring the ground, how to fuse data from different sources?
- We now elaborate and explain this in section 2.4.
- Results: I suggest that you should give an overall description of the experiments.
- This is one of the few studies of this nature (using these methods) focusing on a large carnivore in the neotropics, so there is not much comparable information; thus, these results are unique compared with other studies. Also, it is descriptive in nature, rather than a particular experiment, but its use in an experimental design is something to keep in mind.
- Discussion; I think the significance of the results is not discussed adequately. For example, you should discuss the relationship between the Camera trap and satellite telemetry data, I suggest showing your results in contrast with previous works.
- We add an additional paragraph and highlightthe importance of having satellite telemetry data in long-term study places, especially when abundance is a state variable.
Reviewer 2 Report
see attachment.

Author Response
Please provide the average and range of the distance of the off-trail camera sites to the nearest trail, or the distance to the nearest trail camera site.
- We have provided this information in the methods section.
Details of jaguar individuals’ identification based on images collected by camera trappers is required in the Materials and Methods section. This study deployed one camera per site (Line 86), which means usually they will take picture on either side of a jaguar only, the authors need to explain how to avoid over identify the number of co-existing jaguars especially some individuals (such as number 08 and 09) with only several pictures. Was there any individual identified being different from others based only on pictures of single side? If yes, please explain how to make sure they were different individuals. Over identification of the number of different individuals in the study area will cause over-estimation of their density. Also, which individual was the collared female?
- We have provided this information in the methods section.
Other than genders of the 19 individuals identified in the present study, please also provide their age-group structure, unless all 19 individuals used in analysis were adults.
- During this full-year study we only identified 19 adult individuals using this sample array. As we specified in lines 200-202, we originally defined three classes; cubs (<12 m), young (12-24 m), and adults (>24 months) but we only were able to identify adults during this period.
Figure 3 needs to be re-done; it is difficult to follow individual’s history of existence. Assignment of different color to different individuals was not clear enough at present format, especially the colors between 5 and 10, between 6 and 8, and between 9 and 14 were too like be distinguished. For a better presentation of these information, I will suggest using a timetable with individuals code number on the top row and different months on the first column, and simply fill in the corresponding cell with the same color to show the months when each individual was recorded.
- Figure 3 was fixed.
Details of collared female locations statistic must be provided. Results mentioned in line 194-203 require supporting data. Figure 9 needs more explanations. Also, please be clearer on whether the tracking data of the collared female suggested that the present study over-estimated the density. This is the most valuable contribution by this study.
- We mentioned the 89 km2 home range size (line 331-332) of this female collared individual (Jaguar 03), also citing Montalvo et. al. 2020 [36] as a detailed reference for this data movement. To confirm whether this study over-estimated or not is difficult because we don’t have all the 19 individuals collared, but due the number of individuals from previous years, as well as the seasonal massive resource availability of sea turtles, we are confident these results are realistic. We have clarified in the discussion in lines 433-436 the importance of having telemetry data for density estimates as in this research.
Please add supporting data or results for the recommendations of “population assessments be made at several consecutive 3-month intervals (Line 22)” and “to carry out long-term camera monitoring programs instead of short-term studies to better support jaguar conservation strategies (Line 24-25)”.
- We elaborated this in one more sentence
the recommendation of “intra-camera distance be increased to cover larger areas (Line 23)” may not be appropriate because it most likely will reduce the probability of re-capturing the same individual by cameras, which will further reduce the power of density estimation if using SCR method. This article didn’t provide the data to show the present study area was too small.
- The sampling grid area was 105 km2 and the density state area adjusted with movement data was 160 km2; this area only would embrace 1-2 female jaguars home range (89 km2). Thus our sampling area was small relative to the population of animals we observed and counted, plus considering that males have larger home ranges. So we suggest that if we were to repeat this study, we should increase camera distancing using trails to increase the chances of getting different individuals more times, because we reported lots of recaptures but almost in the same sites.
Reviewer 3 Report
In the manuscript “The use of camera traps and satellite telemetry to estimate jaguar population structure in northwestern Costa Rica”, the authors carried out a camera trap survey for a jaguar population in Costa Rica, estimated the population density, and analyzed the driving factors for detection. I think the data used in this study are solid, yet the presentation of the results needs substantial improvement. I list my comments below:
1. Lines 82-83. “trap nights” was mentioned here. The camera works 24h a day, why used “trap nights”?
2. Figure 1 looks awkward. Panel A and B used the same color/shape combination for the points to represent different things. I suggest to use different shapes of points to represent on-trail and off-trail, different colors to represent different sexes, and use different sizes of the points to represent the number of pictures. The panels B and C can be combined to one panel. In the legend of the figure, use “No.” to replace “Nu.”. “No.” is the most common abbreviation of the word number.
3. In Figure 3, use “No.” to replace “Nu.” In the X axis, change the exact date to month, i.e. change “Jun-16” to “Jun”. Using 19 colors to represent 19 individuals cause confusion. Please use IDs (such as 01, 02) to represent different individuals.
4. The authors used spatial capture-recapture models to explain animal density, detection, and space use, taking into account sessions, sex, and trail. Why not use some ecological meaningful variables such as elevation, vegetation cover, human disturbance, etc.?
5. Figure 7 shows a very interesting pattern, i.e. the satellite-tracked individual moved all around the camera deployed area. But the figure was poorly demonstrated. Please use lines to connect points to show the movement pathways. The figure surprised me a lot. It seems the jaguars’ habitats are largely overlapped. This point deserves further discussion.
6. Figure 8. Change the exact date to month, i.e. change “Jun-16” to “Jun”.
7. Line 223. “For example, [36] found male jaguars are…”. Spell out the author’s name here.
Author Response
Reviewer #3:
Lines 82-83. “trap nights” was mentioned here. The camera works 24h a day, why used “trap nights”?
- With camera trapping studies, the term “trap nights” is widely used to refer to a
continuous 24-hour diel period.
Figure 1 looks awkward. Panel A and B used the same color/shape combination for the points to represent different things. I suggest to use different shapes of points to represent on-trail and off-trail, different colors to represent different sexes, and use different sizes of the points to represent the number of pictures. The panels B and C can be combined to one panel. In the legend of the figure, use “No.” to replace “Nu.”. “No.” is the most common abbreviation of the word number.
- Figure modified as suggested
In Figure 3, use “No.” to replace “Nu.” In the X axis, change the exact date to month, i.e. change “Jun-16” to “Jun”. Using 19 colors to represent 19 individuals cause confusion. Please use IDs (such as 01, 02) to represent different individuals.
- Figure modified as suggested
The authors used spatial capture-recapture models to explain animal density, detection, and space use, taking into account sessions, sex, and trail. Why not use some ecological meaningful variables such as elevation, vegetation cover, human disturbance, etc.?
- We did not include other environmental variables because we focus on abundance and population structure; however, the variables mentioned are useful for further studies of factors affecting occurrence or resource selection at broad scales for jaguars.
Figure 7 shows a very interesting pattern, i.e. the satellite-tracked individual moved all around the camera deployed area. But the figure was poorly demonstrated. Please use lines to connect points to show the movement pathways. The figure surprised me a lot. It seems the jaguars’ habitats are largely overlapped. This point deserves further discussion.
- Figure modified as suggested
Figure 8. Change the exact date to month, i.e. change “Jun-16” to “Jun”.
- Figure modified as suggested
Line 223. “For example, [36] found male jaguars are…”. Spell out the author’s name here.
- Author’s name included
Round 2
Reviewer 2 Report
The authors had addressed all my questions. Several suggestions for consideration:
1. For the estimated density, the article stated “7.7 (Line 210)”, “7.6 (Line 250)” and “~7.7 (Line 291)”, please double check these numbers.
2. In Line 176, please add the calculation or equation for the “radio of captures of a collared female to other jaguars per month”, it is difficult to understand how the Figure 8 was generated.
3. In line 248 “this means that, on average, there were about 2.6 jaguar “units” within the female’s 89-km2 range.” Unless this kind of data can be translated directly into density (need references to support), otherwise I will suggest removing the following sentence “and thus a density of only 2.9 individuals/100 km2 in any one month.” and replaced with a more descriptive sentence for the lower density estimation.
4. I understand and agree that the sampling area should be larger in the future to cover more individuals (as stated in Line 341-342), but I am not fully agreeing to increase the intra-camera distance for covering a larger area (as stated in Line 22-23) unless the distance increasing will NOT reduce the possibility of recapturing the same individual. Instead, I will suggest enlarging the sampling area by increasing number of camera trappers, but not increasing their distance.
5. Sentence in Line 286-289 is difficult to understand.
Author Response
Reviewer 2 – Round 2
Several suggestions for consideration:
For the estimated density, the article stated “7.7 (Line 210)”, “7.6 (Line 250)” and “~7.7 (Line 291)”, please double check these numbers.
-Corrected so all read 7.6
In Line 176, please add the calculation or equation for the “radio of captures of a collared female to other jaguars per month”, it is difficult to understand how the Figure 8 was generated.
-This text/calculation and the figure were deleted in favor of emphasizing the number of individual jaguars detected on the camera grid each month (Fig. 3)
In line 248 “this means that, on average, there were about 2.6 jaguar “units” within the female’s 89-km2 range.” Unless this kind of data can be translated directly into density (need references to support), otherwise I will suggest removing the following sentence “and thus a density of only 2.9 individuals/100 km2 in any one month.” and replaced with a more descriptive sentence for the lower density estimation.
-Deleted – see above
We understand and agree that the sampling area should be larger in the future to cover more individuals (as stated in Line 341-342), but we are not fully agreeing to increase the intra-camera distance for covering a larger area (as stated in Line 22-23) unless the distance increasing will NOT reduce the possibility of recapturing the same individual. Instead, I will suggest enlarging the sampling area by increasing number of camera trappers, but not increasing their distance.
-Agreed – intra-camera distance wording deleted
Sentence in Line 286-289 is difficult to understand.
-This sentence has been deleted.
Reviewer 3 Report
My comments were well addressed, except one point that I need further discussion about the tracked female, who occurred all over the study area of 105 square km. Now I doubt that 19 individuals were detected in such a small area. More evidences for individual recognition should be provided.
Author Response
Reviewer 3 Round 2
I need further discussion about the tracked female, who occurred all over the study area of 105 square km. Now I doubt that 19 individuals were detected in such a small area. More evidences for individual recognition should be provided.
-We have edited and expanded our description of how we identified individuals. We also note that in the study area, we only ever identified an average of 6 individuals in any given month (n = 1-11, Fig. 3), but over the course of a year, 19 animals used the area (likely, in part, due to the unique situation of turtles nesting on the beaches).